# The value of combining individual and small area sociodemographic data for assessing and handling selective participation in cohort studies: Evidence from the Swedish CardioPulmonary bioImage Study

Carl Bonander[1]*, Anton Nilsson[2,3], Jonas Björk[2,4], Anders Blomberg[5], Gunnar Engström[6], Tomas Jernberg[7], Johan Sundström[8,9], Carl Johan Östgren[10], Göran Bergström[11,12], Ulf Strömberg[13]

1 School of Public Health and Community Medicine, Institute of Medicine, Sahlgrenska Academy, University of Gothenburg, Gothenburg, Sweden, 2 Epidemiology, Population Studies and Infrastructures (EPI@LUND), Lund University, Lund, Sweden, 3 Centre for Economic Demography, Lund University, Lund, Sweden, 4 Clinical Studies Sweden, Forum South, Skåne University Hospital, Lund, Sweden, 5 Department of Public Health and Clinical Medicine, Section of Medicine, Umeå University, Umeå, Sweden, 6 Department of Clinical Sciences, Lund University, Malmö, Sweden, 7 Department of Clinical Sciences, Danderyd University Hospital, Karolinska Institutet, Stockholm, Sweden, 8 Department of Medical Sciences, Clinical Epidemiology, Uppsala University, Uppsala, Sweden, 9 The George Institute for Global Health, University of New South Wales, Sydney, Australia, 10 Department of Health, Medicine and Caring Sciences, Linköping University, Linköping, Sweden, 11 Department of Molecular and Clinical Medicine, Institute of Medicine, Sahlgrenska Academy, University of Gothenburg, Gothenburg, Sweden, 12 Department of Clinical Physiology, Sahlgrenska University Hospital, Region Västra Götaland, Gothenburg, Sweden, 13 Department of Research and Development, Region Halland, Halmstad, Sweden

* carl.bonander@gu.se

## Abstract

### Objectives

To study the value of combining individual- and neighborhood-level sociodemographic data to predict study participation and assess the effects of baseline selection on the distribution of metabolic risk factors and lifestyle factors in the Swedish CardioPulmonary bioImage Study (SCAPIS).

### Methods

We linked sociodemographic register data to SCAPIS participants (n = 30,154, ages: 50–64 years) and a random sample of the study's target population (n = 59,909). We assessed the classification ability of participation models based on individual-level data, neighborhood-level data, and combinations of both. Standardized mean differences (SMD) were used to examine how reweighting the sample to match the population affected the averages of 32 cardiopulmonary risk factors at baseline. Absolute SMDs >0.10 were considered meaningful.

**Data Availability Statement:** The Regional Ethics Committee in Umeå has approved the study

according to the Swedish Ethical Review Act (2003:460) regarding the ethical review of research involving humans (diary number 2010-228-31M, with addendum 2011-02-21, for SCAPIS and 2016-511-31 for the linkage of register data to SCAPIS participants). As dictated by the ethical body that approved the study and the promise to participants in their informed consent, the research data collected in the present study cannot be shared publicly as the data contain potentially identifying and sensitive personal data according to article 9 the General Data Protection Regulation (EU 2016/679), and public availability would compromise participant privacy. The General Data Protection Regulation (EU 2016/679) also classifies de-identified versions of sensitive data that are sufficiently detailed to allow for re-identification as sensitive personal information. According to Swedish law (Law 2003:460 for ethical review of research involving humans), ethical permission is required to process such data. In accordance with Swedish legislation, the data can and will be made available to researchers who meet the criteria for access to confidential data, which includes obtaining their own ethics approval from the Swedish Ethical Review Authority (email: registrator@etikprovning.se; website: https://etikprovningsmyndigheten.se). Data applications can then be made by contacting SCAPIS (email: scapis@scapis.org; website: https://www.scapis.org/data-access/).

**Funding:** The study presented in this paper was funded by research grants from Swedish Research Council for Health, Working life and Welfare (Forte, www.forte.se, grant no. 2017-00414; 2020-00962) and the Swedish Research Council (VR, www.vetenskapsradet.se, grant no. 2019-00198). SCAPIS also received external funding from Swedish Heart-Lung Foundation (www.hjart-lungfonden.se, grant no. not available), Knut and Alice Wallenberg Foundation (www.kaw.wallenberg.org, grant no. 2014-0047), Swedish Research Council (www.vetenskapsradet.se, grant no. 822-2013-2000) and VINNOVA (Sweden's Innovation agency, www.vinnova.se, grant no. 2012-04476), and internal funding from University of Gothenburg and Sahlgrenska University Hospital, Karolinska Institutet and Stockholm county council, Linköping University and University Hospital, Lund University and Skåne University Hospital, Umeå University and University Hospital, Uppsala University and University Hospital (grant numbers not applicable for internal sources of funding). The funders had no role in study design, data collection and analysis, decision to publish, or preparation of the manuscript.

## Results

Combining both individual-level and neighborhood-level data gave rise to a model with better classification ability (AUC: 71.3%) than models with only individual-level (AUC: 66.9%) or neighborhood-level data (AUC: 65.5%). We observed a greater change in the distribution of risk factors when we reweighted the participants using both individual and area data. The only meaningful change was related to the (self-reported) frequency of alcohol consumption, which appears to be higher in the SCAPIS sample than in the population. The remaining risk factors did not change meaningfully.

## Conclusions

Both individual- and neighborhood-level characteristics are informative in assessing study selection effects. Future analyses of cardiopulmonary outcomes in the SCAPIS cohort can benefit from our study, though the average impact of selection on risk factor distributions at baseline appears small.

## Introduction

Selective participation is a general concern in population-based research that aims to make inferences about health outcomes or exposure effects in the general population [1]. For instance, the population-based Swedish CardioPulmonary bioImage Study (SCAPIS, www.scapis.org) aims to improve risk prediction of cardiopulmonary diseases and study disease mechanisms in a general middle-aged population [2]. The study combines new imaging techniques with advanced large-scale 'omics' and epidemiological analyses to characterize a population-based cohort, and is expected to provide new evidence about the prevalence of hidden cardiopulmonary disease and improved prediction models for the general population. To fulfill these aims, the participants of SCAPIS must reflect their intended target population.

However, cohort studies that rely on voluntary clinical examinations tend to be skewed towards healthy individuals with high socioeconomic status [3, 4], and SCAPIS is no exception [5, 6]. This type of non-random participation can pose a severe threat to the internal and external validity of study results [1, 7, 8]. A lack of internal validity implies spurious correlations between exposures (or treatments) and health outcomes [7], and a lack of external validity implies poor generalizability of the study results to the intended target population [1]. These problems may negatively influence the utility of the research findings for public health decision-making [9]. However, they can potentially be remedied by reweighting the study sample to match the intended target population on sociodemographic characteristics using inverse propensity for participation weights [10–13]. Constructing such weights typically requires access to external data on non-participants or (a random sample of) the target population [6, 14].

Population registers with high coverage, such as those available in the Nordic countries, enable linkage of sociodemographic data to each study participant and member of the target population [14]. Such register infrastructures are generally not available in other settings, which presents a challenge for high-quality participation modeling and subsequent adjustment for selective participation. However, previous validation studies have found that using individual-level register data to account for selective participation was able to improve the external validity of study results in some cases [15], but not others [12], indicating that important

**Competing interests:** The authors have declared that no competing interests exist.

differences may remain even with access to rich individual-level data on patient histories and sociodemographics.

Collecting neighborhood-level data on the population may serve as a more practical alternative while retaining a relatively high precision in settings where individual-level data cannot be accessed [16, 17]. While neighborhood data alone cannot fully capture and adjust for individual-level selection effects [16], it may also encode contextual influences on participation, risk factors, and health outcomes [18]. Combining data on individual sociodemographic profiles and neighborhood conditions may help account for selection effects beyond those that individual- or neighborhood-level data can account for separately. To our knowledge, only one previous study has directly compared the use of individual-level and aggregate data for handling selective participation; that study focused on statistical approaches that use aggregate summary-level statistics and compared them to a gold-standard individual-level approach [17]. The findings suggested that while individual-level data are preferable, aggregate data can also be leveraged to improve external validity. However, the focus of that study was not on combining data at both levels, and it did not consider neighborhood-level data at a fine scale.

The Swedish register infrastructure, which contains rich information on the entire population [19], provides a useful setting for evaluating the use of individual and area-level data for improving the external validity of study results, both on their own and in combination. The objective of the present study was to investigate the value of combining individual-level and area-level sociodemographic register data for predicting study participation in the context of the Swedish register infrastructure, using the Swedish CardioPulmonary bioImage Study (SCAPIS) as a case study. We also applied the method to assess the potential effects of baseline selection on the distribution of metabolic risk factors and lifestyle factors, which will help inform future research about potential biases caused by selective participation in SCAPIS.

## Methods

### Recruitment and participation in SCAPIS

The recruitment strategy and overall design of the SCAPIS cohort are documented in detail elsewhere [2] and will only be briefly summarized here. To recruit study participants for SCAPIS, written invitations were sent to 59,909 randomly selected men and women between 50 and 64 years of age living in the areas surrounding six university hospitals in Sweden. Of the invited individuals, 30,154 (50.3%) agreed to participate, and baseline clinical examinations were completed between 2013 and 2018. Site-specific details are provided in Table 1.

### Data collection

The present study combines external individual- and neighborhood-level register data on the participants of SCAPIS with data from a random sample of the target population living in the same areas, at the same time, as those invited to participate in the cohort study (Table 1). Specifically, the target population consists of individuals aged 50 to 64 years living in one of the 1,925 demographic statistical areas (DeSO [In Swedish: *Demografiska statistikområden*]) surrounding the university hospitals that were included in SCAPIS (out of the 5,984 DeSOs in Sweden; see the map in S1 Fig in S1 Appendix for reference) sometime between 2013 and 2018 depending on site (see Table 1 for details). The DeSO geography is one of the finer geographical divisions available in Sweden. It was created by Statistics Sweden with the intention of monitoring segregation and socioeconomic conditions in small areas, which makes it especially useful for capturing variation in socioeconomic deprivation at the area level [20]. Throughout the study period (2013–2018), approximately 280 individuals aged 50 to 64 years lived in an average DeSO within the study area (range: 2–546; interquartile range: 229–333).

**Table 1. Participation and recruitment into the Swedish CardioPulmonary bioImage Study (SCAPIS) by site and in total.**

| Site | Population size 50–64 years, n[a] | Randomly invited to participate, n (% of age-matched population) | SCAPIS participants, n (% of invited) | Recruitment period |
|---|---|---|---|---|
| Gothenburg | 90,782 | 12,109 (13.3%) | 6,265 (51.7%) | 2013–2017 |
| Malmö/ Lund | 51,667 | 11,763 (22.8%) | 6,251 (53.1%) | 2014–2018 |
| Linköping | 25,611 | 8,721 (34.1%) | 5,057 (58.0%) | 2015–2018 |
| Stockholm | 331,681 | 11,950 (3.6%) | 5,038 (42.2%) | 2015–2018 |
| Uppsala | 35,242 | 10,763 (30.5%) | 5,036 (46.8%) | 2015–2018 |
| Umeå | 25,659 | 4,603 (17.9%) | 2,507 (54.5%) | 2015–2018 |
| Total | 560,642 | 59,909 (10.7%) | 30,154 (50.3%) | 2013–2018 |

[a] Averaged over the recruitment period within each site.

To simplify the presentation, we will refer to DeSOs as neighborhoods throughout the rest of the paper.

The Swedish Total Population Register covers the entire Swedish population and includes information such as age, country of birth, and place of residence [21]. Based on this register, Statistics Sweden provided anonymized data on the study area population from 2013 to 2018, from which we drew a random sample of individuals (n = 59,909) to represent the target population of SCAPIS. The sample was drawn with the same sampling probabilities from the same neighborhoods and recruitment periods as those invited to participate in SCAPIS (Table 1) and therefore represents the same target population as those invited to participate in the study [14]. For each member of the target population, we also received individual-level data on income divided into three groups based on household disposable income per consumption unit ('low', income in the lowest quartile of the households in Sweden; 'medium', income in quartiles 2 and 3; and 'high', income in the highest quartile) and immigrant group (three groups according to country of birth: the Nordic countries, other Western countries, and non-Western countries, the latter referring to inhabitants born in Eastern Europe, Asia, Africa or South America). Statistics Sweden also linked corresponding data to the SCAPIS participants via Swedish personal identification numbers [22]. The individual-level variables were derived from the Income and Taxation Register and the Total Population Register, which contain data on all Swedish taxpayers and the entire population, respectively.

In addition to the individual-level information, we also linked data on neighborhood-level aggregates of the above-mentioned income and immigrant groups (percentages of the population aged 50–64 years in each group) to each individual, in addition to the following indicators of neighborhood socioeconomic conditions obtained from Statistics Sweden: the percentage of individuals aged 50–64 years with a university degree, the percentage of unemployed working-age individuals, the percentage of single-parent households (all ages), and the percentage of the population living in rental housing (all ages).

## Statistical analysis

**Estimation of propensity scores for participation.** We used multivariable logistic regressions to model participation in a stacked dataset containing both the participants and the population sample (n = 30,154+59,909 = 90,063). With this data structure, the estimated odds of belonging to the participant sample in the data set can be interpreted as the estimated propensity score (i.e., probability) for participation [14]. We note that a practical issue that may arise with this approach is that the estimated propensity score may sometimes exceed one by

chance; in the few cases when this occurred (0 to 0.12% of observations depending on the model), a score of one was assigned for simplicity. The regression models were estimated in R (version 4.0.4; R Core Team, Vienna, Austria).

**Model assessment and comparison.**   We assessed the classification ability of regression models based on individual-level sociodemographics only, area-level data only, and a combination of data from both levels.

The individual-level model contained the following individual-level characteristics: age, sex, income, and country of birth. The area-level model contained the following neighborhood-level characteristics: site, percentage of households with low and middle income, percentage of the population of non-Nordic and non-Western origin, percentage with a university education, percentage living in rental housing, percentage of unemployed working-age individuals, and the percentage of single-parent households (percentage of high-income households and percentage of Nordic origin were omitted to avoid collinearity with the other income and country of birth categories). The combined model contained all characteristics included in the individual-level and area-level models.

In addition to these models, we also estimated a model with spline terms to assess deviations from non-linearity for continuous variables and a model with two-way interactions between all variables.

We calculated the area under the receiver operating characteristic curve (AUC) using an approach developed for participation modeling with stacked datasets [23] (see S1 Appendix for details). The AUC calculations were performed in Stata (version 16.1; StataCorp LLC, College Station, Texas).

**Assessment of changes in cardiopulmonary risk factors after reweighting.**   We used the combined model with two-way interactions to compute inverse probability for participation weights for the SCAPIS participants. These weights were then used to examine changes in the distribution of 32 cardiopulmonary risk factors (see results section for details; data collection procedures used in SCAPIS are detailed elsewhere [2]).

To facilitate comparison between variables of different scales, we computed standardized differences using methods appropriate for categorical and continuous variables [24, 25], which is the recommended approach for assessing covariate balance between groups (e.g., a sample and a population) and between unweighted and weighted samples [24]. An absolute standardized difference above 0.10 is typically used as a reference point to indicate a meaningful covariate imbalance [26], where 0.10 can be read as 10% of one standard deviation of the variable in question.

## Ethics statement

This project has been approved by the Regional Ethics Committee in Umeå (diary number 2010-228-31M, with addendum 2011-02-21, for SCAPIS and diary number 2016-511-31 for the linkage of register data to SCAPIS participants). Written informed consent was obtained from all SCAPIS participants.

Statistics Sweden delivered the population data to us in aggregate form (the individual-level information was recreated from stratified counts). These data cannot be linked to any living person and does not constitute sensitive personal data, and their use is therefore exempt from the need for ethics approval according to the Swedish Ethical Review Act (2003:460).

## Results

The standardized differences between SCAPIS participants and the target population exceeded a magnitude of 0.10 in 12 out of 15 sociodemographic characteristics (Table 2). The most

**Table 2.** Sociodemographic characteristics of the participants in the Swedish CardioPulmonary bioImage Study (SCAPIS), a random sample of its target population, and inferred characteristics of the non-participants of the study.

| Characteristic | Participants | Target population (sample) | Absolute SMD[a] | Non-participants[b] |
|---|---|---|---|---|
| **n** | 30,154 | 59,909 | | 29,755 |
| **Men, n (%)** | 14,646 (48.6) | 29,822 (49.8) | 0.024 | 15,180 (51.0) |
| **Age group, n (%)** | | | 0.080 | |
| 50–54 y | 10,049 (33.3) | 22,000 (36.7) | | 11,945 (40.1) |
| 55–59 y | 9,980 (33.1) | 19,693 (32.9) | | 9,729 (32.7) |
| 60–64 y | 10,125 (33.6) | 18,216 (30.4) | | 8,081 (27.2) |
| **Income group, n (%)** | | | 0.291 | |
| High | 16,927 (56.1) | 26,980 (45.0) | | 10,043 (33.8) |
| Middle | 10,630 (35.3) | 22,636 (37.8) | | 12,001 (40.3) |
| Low | 2,597 (8.6) | 10,293 (17.2) | | 7,711 (25.9) |
| **Country of birth, n (%)** | | | 0.244 | |
| Nordic | 26,074 (86.5) | 46,367 (77.4) | | 20,286 (68.2) |
| Other western | 601 (2.0) | 1,396 (2.3) | | 775 (2.6) |
| Non-western | 3,479 (11.5) | 12,146 (20.3) | | 8,694 (29.2) |
| **Site, n (%)** | | | 0.105 | |
| Gothenburg | 6,266 (20.8) | 12,109 (20.2) | | 5,844 (19.6) |
| Linköping | 5,056 (16.8) | 8,721 (14.6) | | 3,664 (12.4) |
| Malmö | 6,251 (20.7) | 11,763 (19.6) | | 5,512 (18.5) |
| Stockholm | 5,038 (16.7) | 11,950 (19.9) | | 6,912 (23.1) |
| Umeå | 2,507 (8.3) | 4,603 (7.7) | | 2,096 (7.1) |
| Uppsala | 5,036 (16.7) | 10,763 (18.0) | | 5,727 (19.3) |
| **Neighborhood-level characteristics (mean (SD))** | | | | |
| % low income households, ages 50–64 | 13.87 (11.32) | 17.04 (14.36) | 0.245 | 20.3 |
| % middle income households, ages 50–64 | 36.74 (9.85) | 37.92 (9.93) | 0.120 | 39.1 |
| % high income households, ages 50–64 | 49.39 (18.06) | 45.03 (20.24) | 0.227 | 40.6 |
| % of Nordic origin, ages 50–64 | 80.93 (16.95) | 76.35 (21.31) | 0.238 | 71.7 |
| % of other Western origin, ages 50–64 | 2.32 (1.34) | 2.37 (1.41) | 0.038 | 2.4 |
| % of non-Western origin, ages 50–64 | 16.75 (16.76) | 21.29 (21.19) | 0.237 | 25.9 |
| % with university education, ages 50–64 | 45.24 (15.29) | 42.75 (15.85) | 0.160 | 40.2 |
| % unemployed working-age individuals | 20.38 (9.44) | 22.54 (11.28) | 0.208 | 24.7 |
| % single parent households | 6.87 (2.63) | 7.41 (3.04) | 0.190 | 8.0 |
| % rental housing | 28.84 (28.82) | 33.92 (32.06) | 0.167 | 39.1 |

[a] Absolute standardized difference between participants and the target population sample. P-values are less than 0.001 for all differences.

[b] Inferred using the laws of total expectation and total probability (see Online Supplement for derivations). The numbers for continuous characteristics are estimated means; standard deviations (SD) were not inferred for non-participants.

considerable differences were related to income and country of birth at both the individual and neighborhood levels, followed by neighborhood-level unemployment, single-parent households, rental housing, and university education (Table 2). As determined by these characteristics, SCAPIS participants appeared to have higher individual socioeconomic status and live in more affluent neighborhoods than the target population and non-participants of the SCAPIS study. We also note that the SCAPIS participants were, on average, slightly older than the target population (33% in the age range 50–54 years in SCAPIS, 37% in the target population).

The results from the individual-level, neighborhood-level, and combined multivariable logistic regression models for predicting participation in SCAPIS are presented in Table 3. Histograms of the predicted probabilities can be found in S2 Fig in S1 Appendix. The classification ability (AUC) of the models based only on individual-level or neighborhood-level characteristics were 66.9% and 65.5%, respectively. Combining characteristics from both levels improved the classification ability (AUC: 70.2%). Notably, both individual and neighborhood-level socioeconomic conditions independently predicted participation in the combined model (for instance, the neighborhood percentage of low-income individuals was predictive of participation in SCAPIS even when adjusting for income at the individual level) (Table 3). Including interactions between all included variables and cubic splines to account for potential non-linearity in continuous variables only marginally improved the model's classification ability (AUC: 71.1%; 70.3%, respectively). Predicted probabilities from the combined model with interactions varied considerably within strata defined by the individual-level characteristics and site (e.g., from almost zero to approximately 30% among 60 to 64-year-old women of non-Western origin with low incomes within the same city [Uppsala]; S3 Fig in S1 Appendix).

Comparisons of absolute standardized differences between SCAPIS participants and the target population before and after weighting using the estimated propensity scores from the individual-level, neighborhood-level, and combined models, are presented in Fig 1 (detailed data can be found in S1-S3 Tables in S1 Appendix). As expected, weights based on individual-level data could not balance neighborhood-level characteristics and vice versa. Balance was achieved on all observed characteristics in the combined model, indicating sufficient overlap in covariate distributions between the sample and population to standardize the participants to match the target population.

We applied the weights from the model with individual factors only, area-level factors only, and the combined model with two-way interactions to study changes in 32 cardiopulmonary risk factors measured at baseline. Standardized differences between the weighted and unweighted SCAPIS participants from each model are presented in Fig 2, where an *increase* (positive difference) suggests that the target population has a higher mean value or prevalence (depending on the type of variable) than the participants, and a *decrease* (negative difference) indicates the opposite. Overall, we find that most risk factors changed more substantially when we reweighted the participants using individual and neighborhood-level sociodemographics than using individual or neighborhood data alone (Fig 2). However, even when using weights based on data from both levels, only one determinant (self-reported frequency of alcohol consumption) decreased by more than 0.10 (25% reported drinking once a month or less in SCAPIS versus an estimated 30% in the target population). The remaining factors changed less meaningfully. Two decreased with a magnitude between 0.05 to 0.10 (alcohol consumption in grams per day [7.11 vs. 6.52 g/day on average] and high-density lipoprotein [HDL] cholesterol levels [1.63 vs. 1.59 mmol/L], and five increased with a magnitude between 0.05 and 0.10 (current smokers [12% vs. 14%], dyspnea [9.5% vs. 11.6%], body mass index [27.0 vs. 27.2 kg/m$^2$], triglyceride levels [1.25 vs. 1.29 mmol/L], and estimated glomerular filtration rate [85.1 vs. 85.9 ml/min/1.73 m$^2$]). The remaining 24 determinants changed with a magnitude of less than 0.05 (Fig 2). These changes were similar within age groups (S4-S6 Figs in S1 Appendix), indicating that the observed changes are not only driven by the difference in age structure between the sample and population and that socioeconomic conditions also play a key role. Detailed descriptive statistics for each determinant before and after weighting based on the combined individual- and neighborhood-level weights can be found in S4 Table in S1 Appendix.

**Table 3. Results from logistic regression models predicting participation in SCAPIS, with coefficients expressed as odds ratios with 95% confidence intervals in parentheses.**

| | Model | | |
|---|---|---|---|
| **Independent variable** | **Individual only** | **Neighborhood only** | **Combined[a]** |
| **Gender (male)** | 0.957 (0.931, 0.984) | | 0.959 (0.932, 0.986) |
| **Age group** | | | |
| 50–54 years | 1 (reference) | | 1 (reference) |
| 55–59 years | 1.082 (1.045, 1.119) | | 1.085 (1.048, 1.122) |
| 60–64 years | 1.171 (1.131, 1.211) | | 1.170 (1.130, 1.211) |
| **Income group** | | | |
| High | 1 (reference) | | 1 (reference) |
| Middle | 0.801 (0.777, 0.826) | | 0.833 (0.807, 0.860) |
| Low | 0.474 (0.451, 0.498) | | 0.524 (0.497, 0.552) |
| **Country of birth** | | | |
| Nordic | 1 (reference) | | 1 (reference) |
| Other Western | 0.827 (0.750, 0.912) | | 0.867 (0.785, 0.956) |
| Non-Western | 0.629 (0.603, 0.656) | | 0.699 (0.667, 0.732) |
| **Site** | | | |
| Gothenburg | | 1 (reference) | 1 (reference) |
| Linköping | | 0.994 (0.946, 1.045) | 0.994 (0.945, 1.045) |
| Malmö | | 1.269 (1.208, 1.333) | 1.272 (1.211, 1.337) |
| Stockholm | | 0.770 (0.734, 0.809) | 0.767 (0.731, 0.806) |
| Umeå | | 0.897 (0.841, 0.957) | 0.902 (0.845, 0.962) |
| Uppsala | | 0.819 (0.781, 0.860) | 0.819 (0.780, 0.860) |
| **Neighborhood-level characteristics[b]** | | | |
| Prop. low-income households | | 0.231 (0.168, 0.316) | 0.454 (0.329, 0.626) |
| Prop. middle-income households | | 1.062 (0.853, 1.322) | 1.200 (0.961, 1.500) |
| Prop. of non-Western origin | | 0.685 (0.576, 0.814) | 0.961 (0.804, 1.150) |
| Prop. of (non-Nordic) Western origin | | 0.394 (0.115, 1.349) | 0.413 (0.119, 1.431) |
| Prop. with university education | | 1.331 (1.136, 1.558) | 1.346 (1.148, 1.577) |
| Prop. rental housing | | 1.082 (0.998, 1.172) | 1.079 (0.995, 1.170) |
| Prop. unemployed working-age individuals | | 0.639 (0.459, 0.890) | 0.569 (0.408, 0.794) |
| Prop. single-parent households | | 0.598 (0.295, 1.210) | 0.607 (0.298, 1.236) |
| **Model diagnostics** | | | |
| Observations | 90,063 | 90,063 | 90,063 |
| Log Likelihood | -56,298.960 | -56,529.840 | -55,945.500 |
| Akaike Inf. Crit. | 112,613.900 | 113,087.700 | 111,933.000 |
| AUC | 0.6692 | 0.6554 | 0.7017 |

[a] Not including interaction terms.

[b] Neighborhood-level characteristics were entered as proportions.

## Discussion

Our study demonstrates the potential usefulness of combining individual-level register data on sociodemographic characteristics with neighborhood-level data to improve the validity of study results in the presence of selective participation. Notably, our combined model showed a comparable classification ability to a participation model developed for the pilot phase of SCAPIS (AUC: 71.1% versus 73.2%) [6], which used considerably more detailed individual-level data on sociodemographic and disease histories.

## (a) Weighted w.r.t. individual factors

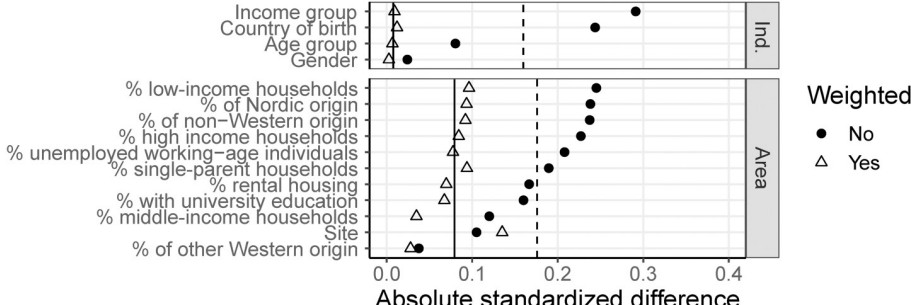

## (b) Weighted w.r.t. neighborhood factors

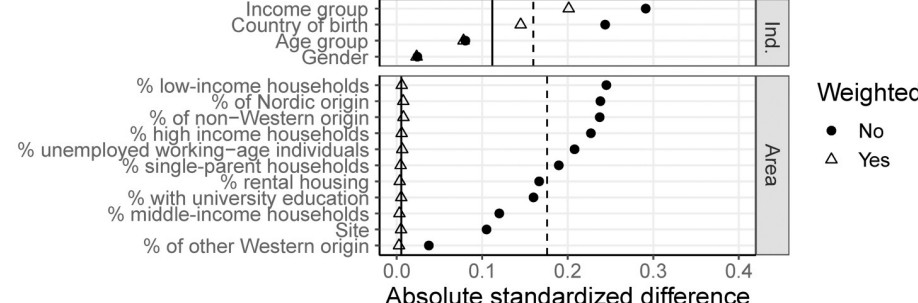

## (c) Weighted w.r.t. individual and neighborhood factors

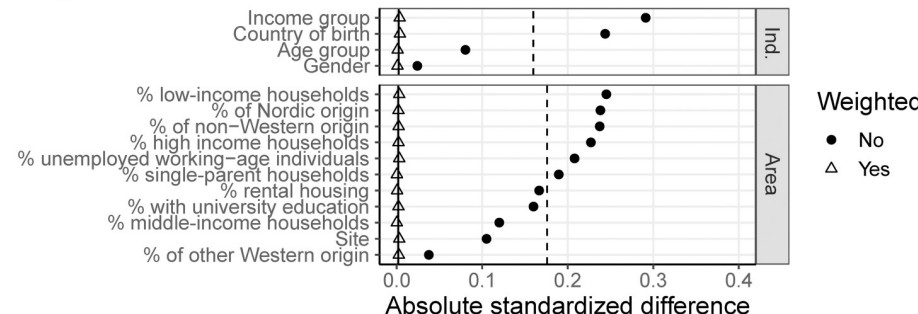

**Fig 1.** Balance (in absolute standardized differences) between SCAPIS participants and the target population before and after inverse probability for participation weighting based on (a) individual characteristics, (b) neighborhood characteristics and (c) the combination of both. The variables are ordered from largest to smallest unweighted standardized difference with variable groups (individual [Ind.] and neighborhood [Area]). The standardized difference, averaged over all included variables before and after weighting within variable groups, are illustrated with dashed and solid vertical lines, respectively.

Together with related research [17], our study provides quantitative insight into the relative importance of individual- and neighborhood-level data in study participation models. Importantly, meaningful differences in income and country of birth remained in our data after adjustment for neighborhood-level characteristics, suggesting that weights based on neighborhood-level data may fail to capture individual-level selection effects. This result highlights a potential problem with using only neighborhood-level data to address selection issues, especially since disease outcomes are more strongly associated with individual-level lifestyle factors than area-level factors [27]. Conversely, another key implication from our study is that the addition of neighborhood characteristics may substantially improve the quality of the participation model even when individual-level data are available. We also found a larger shift in the

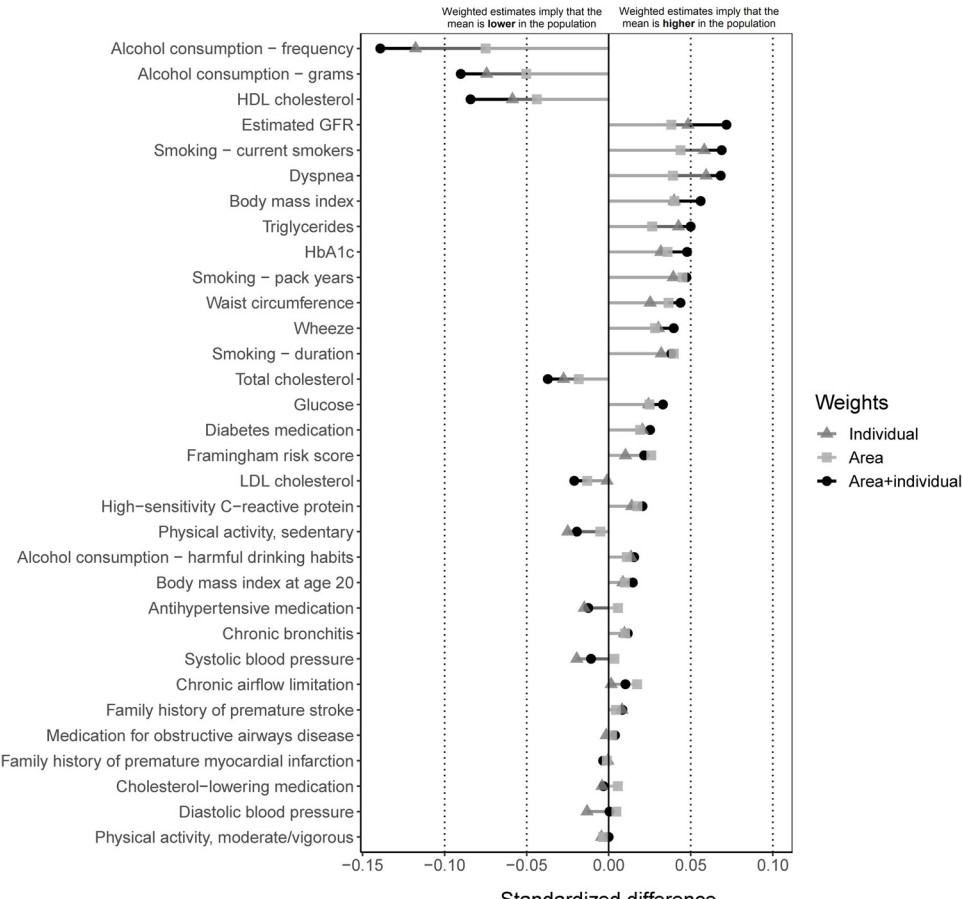

**Fig 2. Standardized differences between the unweighted SCAPIS participants and weighted SCAPIS participants standardized to match the target population on individual and neighborhood-level sociodemographic characteristics.** The horizontal lines show by much the mean changes after reweighting the data using three sets of weights (based on area data only, based on individual data only, or based on both). The vertical reference lines at -0.10, -0.05, 0.05 and 0.10 highlight potentially meaningful differences. An increase in mean (or prevalence, depending on variable type) suggests that the mean is greater in the target population than among SCAPIS participants. A decrease suggests the opposite (i.e., that the estimates incidate a lower mean in the target population relative to SCAPIS participants).

risk factor distribution when using weights based on the combination of individual- and neighborhood-level data than when using weights based on either data source alone. These results imply that one may leverage area-level information to improve the validity of study results even if individual-level data are available. However, additional research is required to assess how these results extend to other contexts and area-level data at other scales (e.g., less fine-scaled geographical units).

The results also have implications for research based on SCAPIS (and similar cohort studies). One important takeaway is that the SCAPIS participants appear reasonably similar to the target population on the distribution of baseline risk factors. If anything, our weighted estimates suggest that the validity of analyses related to self-reported alcohol consumption and, to a lesser extent, renal function, body mass index, dyspnea, and smoking may be affected by selective participation. Specifically, SCAPIS participants appear to consume alcohol more frequently, although the difference could potentially be explained by socioeconomic differences in self-reporting bias [28]. The frequency of alcohol consumption should also not be confused

with a higher prevalence of problem drinking [29], which did not differ as much between the sample and population according to our estimates (Fig 2). We note, however, that a recent study examining the effects of selection in the UK Biobank found that the association between alcohol consumption and cardiovascular disease seems to be particularly affected by sample selection [30].

The prevalence of current smokers also appears to be lower in the SCAPIS sample than in the target population. This measure is also self-reported, but bias in self-reported smoking does not appear to be as sensitive to socioeconomic status as measures of alcohol use [31]. The SCAPIS participants also seem to have a slightly lower body mass index and renal function (measured by estimated glomerular fibrillation rates [32]) and a lower prevalence of dyspnea. Overall, the directions of change in these factors after weighting the sample to match a less advantaged target population are generally in line with our expectations given previous research on their socioeconomic gradients [29, 33–36]. The average SCAPIS participant seems to have higher HDL cholesterol and lower triglyceride levels, suggesting that lipid profiles may differ between SCAPIS participants and the target population. This result can potentially be explained, at least in part, by the negative effect of smoking on HDL [37] and the positive association between BMI and triglycerides [38].

Each of the risk factors listed above is associated with premature mortality and morbidity [39–45], and a meaningful imbalance between the sample and target population could therefore imply a risk of selection bias and lack of generalizability. Such biases depend on the causal pathway(s) between the sample selection mechanism, target parameter, and outcome(s) of interest (see, e.g., references [11, 13, 46] for details). While they, therefore, need to be evaluated before each analysis, our results may provide important clues as to which analyses may be more problematic than others. For instance, SCAPIS researchers may need to proceed with extra caution when analyzing associations between alcohol or smoking habits and cardiopulmonary outcomes.

## Limitations

There are some limitations to our analyses that are important to keep in mind. Firstly, we can only account for selection due to the sociodemographic factors that we observed in our data, but study participation may also depend on other, unobserved factors. Secondly, we currently lack the prospective data required to fully assess the potential bias due to selective participation in associations between cardiopulmonary risk factors and prospective outcomes in SCAPIS. In general, the rich register infrastructures available in the Nordic countries allow for comprehensive investigations into the effects of selection [12, 15], which could be used to extend our analyses once sufficient prospective data become available.

## Conclusions

The accuracy of the SCAPIS participation model was improved by combining individual and small area sociodemographic data. Reweighting the study participants based on this model led to more considerable changes in cardiopulmonary risk factor distributions than using either data source alone. Thus, combining individual and area-level data can potentially improve the assessment and handling of selective participation in cohort studies.

## Supporting information

**S1 Appendix. Supplementary tables, figures and mathematical derivations.**
(DOCX)

## Acknowledgments

We thank the participants and investigators of SCAPIS for enabling this study. We are also grateful for the coordination and assistance provided by Sofia Swedenborg (Swedish Heart-Lung Foundation) to facilitate the writing of this paper.

## Author Contributions

**Conceptualization:** Carl Bonander, Anton Nilsson, Jonas Björk, Göran Bergström, Ulf Strömberg.

**Data curation:** Carl Bonander, Ulf Strömberg.

**Formal analysis:** Carl Bonander, Anton Nilsson.

**Funding acquisition:** Carl Bonander, Anders Blomberg, Gunnar Engström, Tomas Jernberg, Johan Sundström, Carl Johan Östgren, Göran Bergström, Ulf Strömberg.

**Investigation:** Anders Blomberg, Gunnar Engström, Tomas Jernberg, Johan Sundström, Carl Johan Östgren, Göran Bergström, Ulf Strömberg.

**Methodology:** Carl Bonander, Anton Nilsson, Jonas Björk, Ulf Strömberg.

**Project administration:** Carl Bonander, Anders Blomberg, Gunnar Engström, Tomas Jernberg, Johan Sundström, Carl Johan Östgren, Göran Bergström, Ulf Strömberg.

**Resources:** Anders Blomberg, Gunnar Engström, Tomas Jernberg, Johan Sundström, Carl Johan Östgren, Göran Bergström.

**Supervision:** Göran Bergström, Ulf Strömberg.

**Visualization:** Carl Bonander.

**Writing – original draft:** Carl Bonander.

**Writing – review & editing:** Anton Nilsson, Jonas Björk, Anders Blomberg, Gunnar Engström, Tomas Jernberg, Johan Sundström, Carl Johan Östgren, Göran Bergström, Ulf Strömberg.

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
