## [Decision Letter · Decision Letter 0]

14 Dec 2021

PONE-D-21-27261The value of combining individual and small area sociodemographic data for assessing and handling selective participation in cohort studies: evidence from the Swedish CardioPulmonary bioImage StudyPLOS ONE

Dear Dr. Bonander,

Thank you for submitting your manuscript to PLOS ONE. After careful consideration, we feel that it has merit but does not fully meet PLOS ONE’s publication criteria as it currently stands. Therefore, we invite you to submit a revised version of the manuscript that addresses the points raised during the review process.

We look forward to receiving your revised manuscript.

Kind regards,

Dinh-Toi Chu, PhD

Academic Editor

PLOS ONE

“The study presented in this paper was funded by research grants from Swedish Research Council for Health, Working life and Welfare (Forte, grant no. 2017-00414; 2020-00962) and the Swedish Research Council (VR, grant no. 2019-00198). SCAPIS also received external funding from Swedish Heart-Lung Foundation, Knut and Alice Wallenberg Foundation (grant no. 2014-0047), Swedish Research Council (grant no. 822-2013-2000) and VINNOVA (Sweden’s Innovation agency, grant no. 2012-04476), and internal funding from University of Gothenburg and Sahlgrenska University Hospital, Karolinska Institutet and Stockholm county council, Linköping University and University Hospital, Lund University and Skåne University Hospital, Umeå University and University Hospital, Uppsala University and University Hospital (grant numbers not applicable for internal sources of funding). The funders had no role in study design, data collection and analysis, decision to publish, or preparation of the manuscript.”

“The study presented in this paper was funded by research grants from Swedish Research Council for Health, Working life and Welfare (Forte, www.forte.se, grant no. 2017-00414; 2020-00962) and the Swedish Research Council (VR, www.vetenskapsradet.se, grant no. 2019-00198). SCAPIS also received external funding from Swedish Heart-Lung Foundation (www.hjart-lungfonden.se, grant no. not available), Knut and Alice Wallenberg Foundation (www.kaw.wallenberg.org, grant no. 2014-0047), Swedish Research Council (www.vetenskapsradet.se, grant no. 822-2013-2000) and VINNOVA (Sweden’s Innovation agency, www.vinnova.se,  grant no. 2012-04476), and internal funding from University of Gothenburg and Sahlgrenska University Hospital, Karolinska Institutet and Stockholm county council, Linköping University and University Hospital, Lund University and Skåne University Hospital, Umeå University and University Hospital, Uppsala University and University Hospital (grant numbers not applicable for internal sources of funding). The funders had no role in study design, data collection and analysis, decision to publish, or preparation of the manuscript.”

5. Please note that in order to use the direct billing option the corresponding author must be affiliated with the chosen institute. Please either amend your manuscript to change the affiliation or corresponding author, or email us at plosone@plos.org with a request to remove this option.

Reviewers' comments:

Reviewer #1: The manuscript presents a useful approach of improving the accuracy of the SCAPIS participation model by combining individual and small area sociodemographic data. Reweighting the study participants based on this model led to larger changes in cardiopulmonary risk factor distributions than using either data source alone. The combination of individual and area-level data shows a potential improvement of the assessment and handling of selective participation in cohort studies.

Reviewer #2: The authors have addressed an important issue in an intelligible fashion, written in standard English. Moreover, the manuscript is technically sound, and the data support the conclusions. The statistical analysis has been performed appropriately and rigorously.

Reviewer #3: The research presents a new method for evaluating and dealing with selective participation, which could lead to the enhancement of the results obtained, but attention should be paid to the following issues in order to enhance the results of this study:

_ There is a need to rewrite the abstract section according to the way used in POLS ONE, which is in the form of a single unstructured continuous paragraph;

- The researchers did not fully explain why they made restrictions to data availability, although it is possible to provide this data after concealing the information that may lead to the disclosure of the personal identity of the participants;

- The introduction of the draft includes some statistical terms that need further clarification, and no explanatory idea or clarification was given about the SCAPIS study or any other studies that were conducted on this subject. All these issues can help to reach the rationale and justification for doing this study;

- The method section is better to divide it into subsections and explain in a clearer way the method of work. Some other important points should be added such as the method of sampling, data collection, exclusion criteria and the period and duration of the study. It is better to put all tables in the results section. At the end of the method, there should be a subsection for statistical analysis;

- The discussion section did not discuss the results with studies that used the same or comparable method in work, which may affect the generalizability of this method and make it like a pilot project;

- There are some linguistic errors and grammatical problems that affect the structure and context of phrases and sentences. it greatly affected the clarity of the research material.

Reviewer #4: Summary of the research and overall impression

The population-based Swedish Cardio Pulmonary bioImage Study (SCAPIS) is a prospective observational study of a randomly selected sample from the general population and baseline clinical examinations were carried out between 2013 and 2018. The purpose was to improve the risk prediction of cardiopulmonary diseases in the general population by obtaining information on underlying disease mechanisms with a view to better prevention and treatment of CVD, COPD and associated metabolic diseases. This present study is quite an excellent informative one where external individual and neighborhood (small area units) level register data on SCAPIS participants were combined with data from a random sample of their target population to form a stacked dataset. This study finds that using a combination of individual and neighborhood level characteristics improved the accuracy of the participation probabilities for the SCAPIS (predicting participation in SCAPIS) better than using either dataset alone. Furthermore, reweighting the SCAPIS participants using this combined model, led to more marked changes in the cardiopulmonary risk factor distributions at baseline although this change was found to be more meaningful in only one risk factor : self-reported frequency of alcohol consumption.

One of the main strengths of this paper is the inclusion of neighborhood level data at baseline selection of study participants and testing the value of the combined dataset in relation to the individual only and neighborhood only datasets. Thus selection effects at all levels were assessed. Additionally, some of the acknowledged methods for adjusting for selection bias in cohort and other studies were applied in this study. These are inverse probability weighting, controlling for covariates associated with selection and bias analysis.

However, a few clarifications are needed. If these are addressed, I believe the authors would have satisfied the publication criteria for PLOS ONE. These clarifications are the following:

Major areas for improvement

Introduction

1. Is there evidence from literature that a similar study, testing these sets of variables (individual only, neighborhood only and combined) (howbeit for another set of risk factors or other diseases) has been carried out? If so, could the authors please comment on the findings of such studies and if none was found; state so.

Methods

1. The authors did not indicate when this present study was carried out and the duration.

2. The authors might consider mentioning the relationship between one neighborhood in line 99 (where approximately 280 individuals live) and one DeSO in Sweden. Otherwise the following phrase in line 97 to 98 would be confusing- “In this paper, we refer to these area units as neighborhoods” -since the randomly selected target population was way above 280. This is for the benefit of readers not familiar with this Swedish system.

Minor areas for improvement

1. It appears that the following elements were omitted from the model with area level data only in line 126 (but they were mentioned earlier on in the paper): high income, Nordic origin, unemployed working age.

---

## [Author Response · Author response to Decision Letter 0]

25 Jan 2022

Thank you for taking the time to review our manuscript. Please find our detailed response to each comment in the appended file named "Response to reviewers".

---

## [Decision Letter · Decision Letter 1]

23 Feb 2022

The value of combining individual and small area sociodemographic data for assessing and handling selective participation in cohort studies: evidence from the Swedish CardioPulmonary bioImage Study

PONE-D-21-27261R1

Dear Dr. Bonander,

We’re pleased to inform you that your manuscript has been judged scientifically suitable for publication and will be formally accepted for publication once it meets all outstanding technical requirements.

Kind regards,

Dinh-Toi Chu, PhD

Academic Editor

PLOS ONE

---

## [Editor Report · Acceptance letter]

28 Feb 2022

PONE-D-21-27261R1 

The value of combining individual and small area sociodemographic data for assessing and handling selective participation in cohort studies: evidence from the Swedish CardioPulmonary bioImage Study 

Dear Dr. Bonander:

I'm pleased to inform you that your manuscript has been deemed suitable for publication in PLOS ONE. Congratulations! Your manuscript is now with our production department. 

Kind regards, 

on behalf of

Dr. Dinh-Toi Chu 

Academic Editor

PLOS ONE